# Plasma metabolomics profiling identifies new predictive biomarkers for disease severity in COVID-19 patients

Nelson C. Soares[1,2☯], Amal Hussein[3☯], Jibran Sualeh Muhammad[1,4], Mohammad H. Semreen[1,2], Gehad ElGhazali[5], Mawieh Hamad[1,6]*

1 University of Sharjah, Research Institute of Medical and Health Sciences, Sharjah, United Arab Emirates, 2 Department of Medicinal Chemistry University of Sharjah, Department of Medicinal Chemistry, College of Pharmacy, Sharjah, United Arab Emirates, 3 Department of Family and Community Medicine & Behavioral Sciences, College of Medicine, University of Sharjah, Sharjah, United Arab Emirates, 4 Department of Basic Medical Sciences, College of Medicine, University of Sharjah, Sharjah, United Arab Emirates, 5 Department of Immunology, Sheikh Khalifa Medical City- Union71- Purelab, Abu Dhabi and College of Medicine and Health Sciences, United Arab Emirates University, Al Ain, United Arab Emirates, 6 Department of Medical Laboratory Sciences, College of Health Sciences, University of Sharjah, Sharjah, United Arab Emirates

☯ These authors contributed equally to this work.
* mabdelhaq@sharjah.ac.ae

**Data Availability Statement:** The metabolomics data have been deposited in the Metabolomics Workbench repository (https://www.metabolomicsworkbench.org/) with data ID 3469.

## Abstract

Recently, numerous studies have reported on different predictive models of disease severity in COVID-19 patients. Herein, we propose a highly predictive model of disease severity by integrating routine laboratory findings and plasma metabolites including cytosine as a potential biomarker of COVID-19 disease severity. One model was developed and internally validated on the basis of ROC-AUC values. The predictive accuracy of the model was 0.996 (95% CI: 0.989 to 1.000) with an optimal cut-off risk score of 3 from among 6 biomarkers including five lab findings (D-dimer, ferritin, neutrophil counts, Hp, and sTfR) and one metabolite (cytosine). The model is of high predictive power, needs a small number of variables that can be acquired at minimal cost and effort, and can be applied independent of non-empirical clinical data. The metabolomics profiling data and the modeling work stemming from it, as presented here, could further explain the cause of COVID-19 disease prognosis and patient management.

## Introduction

The SARS-COV-2 pandemic, which has gripped the world over the last three years, has resulted in more than 530 million reported infections and 6.3 million deaths worldwide so far [https://covid19.who.int/]. The pandemic also resulted in unimaginable suffering to individuals, families, communities and countries across the globe. At its peak, the pandemic stressed healthcare systems in different parts of the globe to their limit, disrupted supply chains, destroyed businesses, resulted in massive unemployment and poverty and a never-seen-before upward re-distribution of wealth; which will collectively continue to impact life on earth for possibly generations to come [1–3]. Although it is arguable whether the pandemic was

**Funding:** This work was supported by research grant CoV19-0305 (MH), seed grant 2001110138, University of Sharjah, UAE. This research is part of the -Human Disease Biomarkers Discovery Research Group-study. The authors wish to acknowledge the generous support of the Research Institute for Medical and Health Sciences, University of Sharjah UAE. The funders had no role in study design, data collection and analysis, decision to publish, or preparation of the manuscript.

**Competing interests:** The authors have declared that no competing interest exist.

foreseen and/or could have been avoided or even better managed; still, the way in which it was handled speaks of utter incompetence and indisputable lack of preparedness at all levels from governments and healthcare authorities all the way to the scientific community [1–3]. Therefore, the world needs to learn its lesson, not only in terms of how to deal with future epidemics and pandemics but also how to study and understand them and how to use cutting edge technologies in such endeavors [1,2,4].

One of the puzzling questions about the COVID-19 pandemic that still lingers is how and why some, seemingly healthy (low risk) individuals, succumbed to the disease while others, possibly of poorer health, recovered and survived [5–7]. Indeed, most people would agree that the worst of this pandemic is, more or less, behind us, but efforts to uncover infection and disease correlates that may have contributed to its outcomes remain timely and needed [5,6]. In this context, there is a real need to develop and test disparate data-integrating approaches and data-based models to understand the various aspects of COVID-19 and to easily and quickly enlist such models in combating future epidemics and pandemics.

Polymerase chain reaction (PCR) testing of the nasopharyngeal swab for the presence of SARS-CoV-2 RNA continues to be the gold standard in identifying infected individuals [8]. Based on global data input, almost 80% of SARS-COV-2 individuals end up with no symptoms to mild-to-moderate symptoms [9]. Serological testing in the form of differential blood count along with inflammatory marker testing has proven partially successful in identifying patients at high risk of disease severity or death. The experience with COVID-19 has demonstrated that testing for serum IL-6, D-dimer, lactate dehydrogenase (LDH) and other analytes is helpful in identifying patients at risk of sever or fatal complications [10]. That said, there is still a need for more specific predictive parameters of severe infection beyond serum ferritin, prothrombin time, and fibrin degradation products (FDP) [11].

Serum/plasma metabolomics profiling using liquid chromatography-mass spectrometry (LC-MS) has proven useful in identifying diagnostic, prognostic and therapeutic biomarkers in infectious diseases. Studies have shown that serum levels of citrate, malate and succinate increase in response to *S. aureus* and *S. pneumonia* infections [12,13]. In a study on COVID-19 intensive care unit patients, the plasma metabolites kynurenine and arginine ratio was reported to be helpful in predicting COVID-19 disease irrespective of age, gender or hospital admission [14]. However, the role of these findings in COVID-19 prognosis remains limited given that only ICU patients were assessed. In another study, the metabolites cytosine and tryptophan-nicotinamide were reported to be moderately sensitive in discriminating COVID-19 patients from healthy individuals [15]. It is predictable that metabolomic changes resulting from SARS-CoV-2 infection could vary widely among patients owing to differences in patient health profiles, vis-à-vis comorbidities, medications, diet, lifestyle, etc. Accordingly, the search for profiles of metabolic biomarkers may provide higher sensitivity and specificity in assessing disease prognosis [16]. In this study, we retrospectively recruited COVID-19 patients with no known comorbidities and divided them into three groups based on disease severity: asymptomatic, mild and severe. We performed LC-MS metabolomics profiling in serum samples of these patients and identified eight predictive biomarkers of COVID-19 disease severity. We then integrated patients' laboratory findings and metabolomics profiles to generate a predictive model of disease severity.

## Material & methods

### Sample collection and processing

In this retrospective cohort study, blood samples were collected from donors who tested positive for COVID-19 and presented with no, mild or severe symptoms between March 20 until

July 17, 2020. Patients were diagnosed with COVID-19 using a nasal swab PCR test and later divided into three groups (asymptomatic, mild, and severe) based on their clinical presentation. Each donor gave a 10 mL blood sample, one half of which was collected in a plain tube and the other half in an EDTA vacutainer. A total of 85 samples were collected (30 COVID-19-positive asymptomatic, 10 COVID-19-positive with mild symptoms, and 45 COVID-19-positive with severe symptoms) for the purpose of this study. COVID-19-positive asymptomatic individuals were identified as a result of the national screening campaigns. Symptomatic COVID-19 patients were classified into mild or severe based on guidelines published by Abu Dhabi Department of Health (circular number 33, 19[th] April 2020). Patients with mild disease presented with upper respiratory tract infection and symptoms like fever, dry cough, sore throat, runny nose, muscle and joint pains without shortness of breath. Patients with severe disease presented with severe pneumonia and symptoms like fever, cough, dyspnea and fast breathing (>30 per minute), in addition to oxygen saturation <90%. Patient records showed that many of the patients with symptoms were self-medicating with aspirin prior to their hospital visit and that some of the patients with moderate-severe symptoms were placed on dexamethasone and/or heparin subsequent to hospital admission. Immediately upon sample collection, the hospital laboratory staff separated and tested the serum for CRP, D-dimer, ferritin, IL-6 and LDH; a complete blood count was also performed on each sample. Whole blood samples were also aliquoted and frozen at −80 ˚C for subsequent processing and analysis. The study was jointly approved by the Ministry of Health, Abu Dhabi and Dubai Health Authority (DOH/CVDC/2020/1949) on the understanding that samples will be number-coded to hide patient's identity, that no personal information will be shared with a third party and that no sample analysis can be performed by entities other than the Research Institute of Medical and Health Sciences (RIMHS), the University of Sharjah (UOS) without prior written approval. No informed consent was required as per the ethical approval decision (*DOH/CVDC/2020/1949*); in compliance with the said decision, all samples were fully anonymized before accessing or receiving them.

## Serum levels of hepcidin and soluble transferrin receptor (sTfR), sCD163 and haptoglobin (Hp) concentration and phenotype distribution

Upon receipt of frozen samples at RIMHS, UOS laboratories, whole blood samples were thawed and centrifuged; serum was separated and levels of hepcidin (Cat No.733228; MyBiosource, San Diego, California, United States), soluble sTfR (Cat No. 750294; MyBiosource), sCD163 (MBS508555) and Hp (MBS763395) were measured using commercially-available colorimetric assay kits; absorbance was read at 450 nm on a microplate reader. Hp phenotypes were determined by vertical polyacrylamide gel electrophoresis, and the bands were visualized by staining with benzedine solution as previously described [17].

## Liquid chromatography tandem mass spectrometry (LC-MS/MS)

Plasma was obtained after the collection of samples into heparinized tubes followed by centrifugation for 5 minutes (3000*g*). The samples were stored at −80 ºC for long-term storage until further metabolomics analysis. An aliquot of plasma sample was first placed into a microcentrifuge tube where cold methanol was added into the sample at 3:1 v/v (i.e., 30 μL sample, add 90 μL cold methanol) vortex and was then stored at −20ºC for two hrs. Next, the samples were centrifuged at 20,817 x g for 15 min at 4ºC. Then, the supernatant was transferred to a new microcentrifuge tube. Usually, the original sample volume is transferred three times (i.e., for 30 μL sample, add 90 μL cold methanol, then transfer 90 μL supernatant). The sample was dried down using Speed vac at 30–40˚C. The dried sample was then either stored in a −80ºC

freezer for further use or dissolved in solvent for LC-MS/MS analysis. Metabolites were analyzed by HPLC-Q-TOF MS/MS using the auto MS/MS positive scan mode as per described in our recent publications [18,19]. Briefly, samples were chromatographically separated using a Hamilton[R] Intensity Solo 2 C18 column (100 mm x 2.1 mm x 1.8 μm) and eluted using 0.1% formic acid in water (A) and 0.1% formic acid in ACN (B) using the following gradient: at a flow rate of 0.250 mL/min 1% B from 0–2 min, then gradient elution to 99% B from 2–17 min, held at 99% B from 17–20 min, then re-equilibrated to 1% B from 20–30 min using a flow rate of 0.350 mL/min. The autosampler temperature was set at 8˚C and the column oven temperature at 35˚C. The ESI source with dry nitrogen gas was 10 L/min, and the drying temperature was equal to 220˚C with nebulizer gas pressure set to 2.2 bar. The capillary voltage of the ESI was 4500 V and the Plate Offset 500 V. MS acquisition scan was set at 20–1300 m/z and the collision energy at 7 eV. Sodium formate was injected as an external calibrant between 0.1 and 0.3 minutes. A total volume of 10 μL sample was injected into the TIMS-TOF MS.

Processing analysis was performed using MetaboScape[R] 4.0 (Bruker Daltonics). Analyte bucketing and identification were done using the software provided available T-ReX 2D/3D workflow with the following parameters: intensity threshold greater than 1000 counts and peak length equal to 7 spectra or greater. Feature quantitation performed using peak area, for features present in at least 3 (of 12) samples (per cell type) were considered for statistical analysis. Analyte $MS^2$ spectra were averaged on import and only features eluting between 0.3 and 25 min with mz between 50 and 1000. For metabolite identification, both $MS^2$ spectra and retention time (RT) were used with the MS/MS spectra as the minimum criterion for a positive hit. For the set of compounds meeting this criterion (either MS/MS alone or MS/MS with RT), annotation using Bruker's implementation of the Human Metabolome Database (HMDB-4.0) was performed; all selected compounds were matched against this library. Where a particular database entry was matched by multiple features, putatively matching features were filtered by considering each of the features against the highest annotation quality score (AQ score) among other putative matches for the same metabolite; i.e. features exhibiting the best fit across the greatest number of factors such as retention time, MS/MS, m/z values, analyte list and spectral library matching were ranked first for the associated identification as per previous publication [18,19]. Pathway enrichment analyses were performed using MetaboAnalyst V5 (https://www.metaboanalyst.ca). Pathway enrichment evaluates overall pathway impact by considering the relative importance of altered metabolite based on their position in the pathway map.

All data, including raw files, have been deposited in the Metabolomics Workbench Repository (https://www.metabolomicsworkbench.org/).

## Data analysis

The metabolomics data were first tabulated in Microsoft excel format and then exported to the Statistical Package for Social Sciences (SPSS) software, version 27 [20]. Demographics, clinical and metabolites data were all merged into one SPSS dataset. Descriptive statistics was used to conduct univariate analysis; frequencies and relative frequencies were used to condense categorical data while measures of central tendency were performed for scale data. Normality of scale data was first tested graphically, using Q-Q plots and histograms and then statistically analyzed using the Kolmogorov Smirnov test. Mean and standard deviations (SD) were reported for scale variables showing normally distributed data, whereas median and interquartile range (IQR: Q1-Q3) were used to summarize variables with skewed data. Chi-square test was performed to test for associations between categorical variables where the strength of an association was measured using the odds ratio (OR). To study the relationships between a

normally distributed outcome and a categorical dichotomous predictor, the independent t-test was used. If the predictor had more than two groups, one-way ANOVA test was used to compare three or more means. For skewed outcome variables, similar analyses were conducted using the non-parametric tests Mann-Whitney U test and Kruskal-Wallis test, respectively. Spearman correlation coefficient was performed to investigate the correlation between two variables with skewed scale data. P-values less than or equal to 0.05 indicated statistical significance. Bonferroni correction was used to adjust p-values in pairwise comparisons.

The receiver operating characteristic curve (ROC) and the area under the curve (AUC) were performed to identify, from among the clinical and metabolite tests, significant diagnostic biomarkers for predicting the severity of COVID-19 infection. An ROC AUC value above 0.70 indicated moderate to high level of accuracy of prediction. For each test's AUC value, statistical significance was assessed against chance by calculating its 95% Confidence Interval (CI) and associated p-value. For each significant diagnostic test/biomarker showing high/moderate accuracy prediction level (AUC > 0.70), data-driven approach was used to determine the optimal cut-off value, specifically, by maximizing the Youden's index (Sensitivity + Specificity– 1) [21]. Next, the sensitivity (SN) and specificity (SP), along with their 95% confidence intervals, were calculated for each diagnostic test.

Optimal cut-off values were used to dichotomize each biomarker into low and high levels. A low level was coded as zero and a high level was coded as 1. After excluding biomarkers that were linearly related, predictors were identified to develop a risk scoring system to define a diagnostic model for COVID-19 severity based on a combination of important biomarkers used as predictors. The risk score was calculated by counting, for each patient, the number of biomarkers that were of high levels. The ROC and the AUC, using Youden's Index, were then used to identify the optimal risk score for predicting the severity of COVID-19. Demographics, clinical and serum metabolite laboratory test results were first compared between the three levels of COVID-19 infection severity (asymptomatic, mild and severe). Preliminary analysis has shown that the asymptomatic and mild groups were comparable on most clinical and metabolite results; no significant differences were observed between the two groups. Accordingly, the two groups (asymptomatic and mild) were clustered into a single group (asymptomatic/mild), which was then compared to the severe COVID-19 group to conduct the analysis reported in this manuscript.

## Results

### Study population demographics

In this study, we analyzed data pertaining to a total of 85 COVID-19 cases (Fig 1), of whom 35.3% (n = 30) were asymptomatic, 11.8% (n = 10) were mild and 52.9% (n = 45) were severe. Males constituted the majority of patients in this study (84.7%, n = 72) as compared to (15.3%, n = 13) females. Mean age of patients was 42 years (SD = 7.73) with a minimum age value of 27 years and a maximum of 62 years. Age was categorized into two groups where 43.5% (n = 37) were 40 years or younger and 56.5% (n = 48) were older than 40 years (Table 1). Age group and gender distributions were comparable between the two groups (asymptomatic/ mild) vs. severe (Table 1).

### Laboratory findings (clinical tests) and severity of COVID-19 disease

In the study sample as a whole, inflammatory markers including the C-reactive protein (CRP) and D-dimer had median values of 18.2 mg/L (Q1-Q3: 5.45–115.49) and 0.60 μg/ml (Q1-Q3: 0.31–2.24) respectively and mean values of lymphocyte and neutrophil counts of 1.45 (SD = 0.72) and 7.56 (SD = 4.13) cells/μL respectively (Table 1). The majority of the

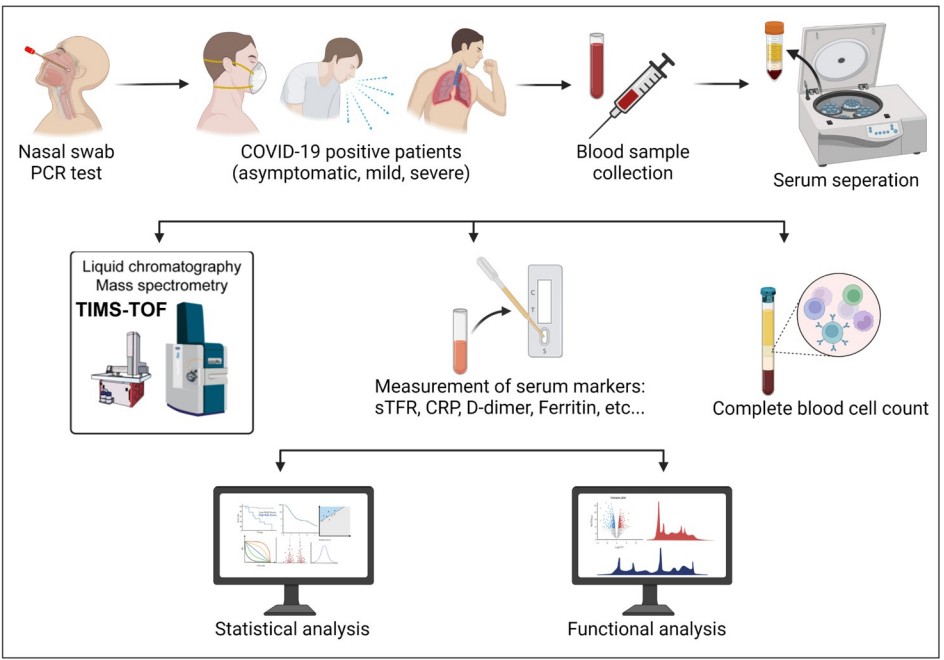

**Fig 1. Graphical figure summarizing the workflow followed in this study.**

inflammatory markers were found to be significantly higher in the severe COVID-19 group relative to the asymptomatic/mild group. For example, CRP had a median value of 5.90 mg/L in the asymptomatic/mild group and 131.22 mg/L in the severe group (U = 1627.50, p-value<0.001). Similarly, D-dimer had median values of 0.28 μg/ml in the asymptomatic/mild group and 1.27 μg/ml in the severe group (U = 161050, p-value<0.001). While lymphocytes were significantly higher in the asymptomatic/mild group (mean = 1.94 cells/μL) than in the severe group (mean = 1.00 cells/μL; t = 7.880, p-value<0.001), neutrophils were significantly higher in the severe (mean = 9.35 cells/μL) as compared to the asymptomatic/mild group (mean = 5.58 cells/μL; t = -4.773, p-value<0.001) (Table 1). No significant differences were found between the asymptomatic/mild group *vs*. the severe COVID-19 group, vis-à-vis median values of serum hepcidin or sCD163. However, Hp and soluble sTfR levels were significantly higher (p-value<0.001) in the severe *vs*. the asymptomatic/mild group; Hp median values were 138.02 *vs*. 115.73 ng/ml and sTfR median values were 31.61 *vs*. 21.46 ng/ml (Table 1).

## Metabolomics profiles of COVID-19 patients according to disease severity

To investigate the possibility of identifying serum metabolites that help in studying the prognosis of disease severity, metabolomics profiling of plasma samples from patients with no, mild and severe symptoms was performed. A total of 99 metabolites were measured and compared between the asymptomatic, mild and severe cases. Pairwise comparisons showed comparable results for asymptomatic *vs*. mild patients, hence the grouping of data obtained from asymptomatic patients and patients with mild symptoms as one "asymptomatic/mild" group; much the same as was done in the previous section. Out of the 99 metabolites (S1 Table), 68 have shown significantly different median values between the asymptomatic/mild group and the severe groups. Of these 68, eight (8) metabolites (K_4_Aminophenol, Acetaminophen glucuronide, Cytosine, Elaidic acid, Glycine, Isobutyric, Paracetamol sulfate and

**Table 1. Demographic and clinical characteristics of Covid-19 patients by severity of Covid-19 (N = 85).**

| Variable | | N | % | Severity of Covid-19 | | Test statistic | p-value |
|---|---|---|---|---|---|---|---|
| | | | | Asymptomatic/ Mild n(%) | Severe n (%) | | |
| Age (years) | Mean (SD) | 42.0 (7.73) | | | | | |
| | Minimum | 27 | | | | | |
| | Maximum | 62 | | | | | |
| Age | < = 40 | 37 | 43.5 | 15(40.5%) | 22(59.5%) | 1.117[#] | 0.290 |
| | >40 | 48 | 56.5 | 25(52.1%) | 23(47.9%) | | |
| Gender | Male | 72 | 84.7 | 33(45.8%) | 39(54.2%) | 0.284[#] | 0.594 |
| | Female | 13 | 15.3 | 7(53.8%) | 6(46.2%) | | |
| Severity | Asymptomatic | 30 | 35.3 | | | | |
| | Mild | 10 | 11.8 | | | | |
| | Severe | 45 | 52.9 | | | | |
| CRP | Mean (SD) | 69.1 (95.25) | | | | 1627.50[*] | <0.001 |
| | Median (Q1-Q3) | 18.2 (5.45–115.49) | | 5.90 | 131.22 | | |
| D-dimer | Mean (SD) | 8.75 (56.02) | | | | 1610.50[*] | <0.001 |
| | Median (Q1- Q3) | 0.60 (0.31–2.24) | | 0.28 | 1.27 | | |
| Ferritin | Mean (SD) | 1153.57 (2072.06) | | | | 1778.00[*] | <0.001 |
| | Median (Q1- Q3) | 377.0 (143.50–1623.50) | | 137.50 | 1765.00 | | |
| IL6 | Mean (SD) | 294.58 (903.15) | | | | 513.00[*] | <0.001 |
| | Median (Q1- Q3) | 57.30 (6.20–144.00) | | 6.20 | 113.00 | | |
| LDH | Mean (SD) | 421.59 (444.45) | | | | 1720.00[*] | <0.001 |
| | Median (Q1- Q3) | 249.0 (159.00–555.00) | | 159.50 | 570.00 | | |
| Lymphocytes | Mean (SD) | 1.45 (0.72) | | 1.94 | 1.00 | 7.880^ | <0.001 |
| | Median (Q1- Q3) | 1.40 (0.82–2.03) | | | | | |
| Neutrophils | Mean (SD) | 7.56 (4.13) | | 5.58 | 9.35 | -4.773^ | <0.001 |
| | Median (Q1- Q3) | 7.28 (4.09–10.22) | | | | | |
| Hepcidin (ng/ml) | Mean (SD) | 3.92 (0.79) | | | | 958.50[*] | 0.606 |
| | Median (Q1- Q3) | 3.71 (3.41–4.24) | | 3.49 | 3.89 | | |
| Haptoglobin (ng/ml) | Mean (SD) | 138.71 (41.85) | | | | 1394.50[*] | <0.001 |
| | Median (Q1- Q3) | 130.26 (110.59–163.61) | | 115.73 | 138.02 | | |
| Transferrin Receptor (ng/mL) | Mean (SD) | 37.27 (41.99) | | | | 1360.00[*] | <0.001 |
| | Median (Q1- Q3) | 26.71 (21.30–37.43) | | 21.46 | 31.61 | | |
| sCD163 (ng/mL) | Mean (SD) | 81.76 (68.59) | | | | 734.00[*] | 0.251 |
| | Median (Q1- Q3) | 66.68 (30.22–113.30) | | 78.82 | 41.15 | | |
| Hp-Hb Phenotype | 1–1 | 7 | 8.2 | | | | |
| | 2–1 | 42 | 49.4 | | | | |
| | 2–2 | 36 | 42.4 | | | | |

[#] Chi-square test;

[*]Mann Whitney U test;

^ Independent t-test.

Succinylacetone) were significantly higher in the severe group. Additionally, sixty (60) metabolites showed significantly higher values in the asymptomatic/mild group *vs.* the severe group (Table 2). Next, we conducted an enrichment analysis of the Biological Process gene ontology terms linked with those metabolites. The enrichment pathway analysis using the "small molecule pathway database (SMPDB)" (available in MetaboAnalyst 5.0 software) revealed that the pathways, that the differentially abundant metabolites were most enriched for included the

**Table 2. Comparing metabolites of patients by COVID-19 disease severity.**

| Variable | Severity of Covid-19 | | Test statistic | p-value |
|---|---|---|---|---|
| | Asymptomatic/Mild | Severe | | |
| K_1_3_Dimethyluric_acid | 2672.50 | 1921.50 | 209.00* | *<0.001* |
| K_1_Methyladenosine | 1663.79 | 1389.44 | 1.711^ | 0.092 |
| K_2_5_Furandicarboxylic_acid | 13377.25 | 9813.51 | 6.994^ | *<0.001* |
| K_2_Pyrrolidinone | 4315.00 | 5268.50 | 813.00* | 0.444 |
| K_3_4_5_Trimethoxycinnamic_acid | 39012.00 | 15228.50 | 406.00* | *<0.001* |
| K_3_5_Dimethoxyphenol | 2725.85 | 2596.80 | 0.715^ | 0.477 |
| K_3_Indolepropionic_acid | 3649.50 | 1395.50 | 299.50* | *<0.001* |
| K_3_Methylindole | 22399.33 | 10513.67 | 13.843^ | *<0.001* |
| K_3_Methylxanthine | 3231.50 | 1296.50 | 171.00* | *<0.001* |
| K_4_Aminophenol | 630.00 | 3825.50 | 1589.00* | *<0.001* |
| K_5_Hydroxy_L_tryptophan | 4799.28 | 1467.60 | 10.005^ | *<0.001* |
| K_5_Hydroxyindoleacetic_acid | 13208.50 | 16004.00 | 694.00* | 0.070 |
| K_9_Methyluric_acid | 3975.00 | 2003.00 | 301.50* | *<0.001* |
| Acetaminophen | 3416.00 | 29426.50 | 1420.00* | *<0.001* |
| Acetaminophen_glucuronide | 630.00 | 8756.00 | 227.00* | *0.013* |
| Acetic_acid | 50286.00 | 55482.00 | 1101.00* | 0.077 |
| Acetone | 37615.50 | 37901.00 | 827.00* | 0.520 |
| Adenosine_monophosphate | 15222.50 | 5947.00 | 288.00* | *<0.001* |
| Allantoin | 1850.15 | 1670.58 | 1.747^ | 0.085 |
| Alpha_ketoisovaleric_acid | 15663.85 | 7053.31 | 13.536^ | *<0.001* |
| Alpha_N_phenylacetul_L_glutamine | 28288.00 | 26065.00 | 944.00* | 0.698 |
| Aniline | 3662.50 | 4178.00 | 739.00* | 0.156 |
| Aspartame | 11128.35 | 11444.58 | -0.307^ | 0.760 |
| Azelaic_acid | 2511.00 | 1644.50 | 470.00* | *<0.001* |
| Benzaldehyde | 5368.25 | 4383.58 | 3.175^ | *0.002* |
| Benzocaine | 2552.35 | 2549.02 | 0.016^ | 0.988 |
| Benzoic_acid | 7121.50 | 4554.00 | 387.00* | *<0.001* |
| Cadaverine | 19435.50 | 14872.00 | 610.00* | *0.011* |
| Caffeine | 310031.00 | 74711.50 | 287.00 | *<0.001* |
| Chlorpheniramine | 3745.35 | 3101.23 | 3.202^ | *0.002* |
| Cinnamic_acid | 27429.55 | 23088.89 | 3.058^ | *0.003* |
| Cis_Aconitic_acid | 4119.38 | 3359.31 | 5.606^ | *<0.001* |
| Cortisol | 8813.00 | 7104.50 | 721.00* | 0.115 |
| Creatine | 4351.00 | 2946.50 | 353.00* | *<0.001* |
| Cytosine | 490.00 | 1170.50 | 1379.00* | *<0.001* |
| Deoxycholic_acid_glycine_conjugate | 13441.00 | 3219.50 | 350.00* | *<0.001* |
| DL_2_aminooctanoic_acid | 3028.50 | 772.00 | 210.50* | *<0.001* |
| Elaidic_acid | 6302.50 | 7884.00 | 1154.00* | *0.025* |
| Ethanolamine | 2644.00 | 2186.00 | 413.00* | *<0.001* |
| Glucosamine | 2291.50 | 1898.00 | 698.00* | 0.103 |
| Glycerophosphocholine | 2410.00 | 1275.00 | 157.00* | *<0.001* |
| Glycine | 559.00 | 735.00 | 1152.00* | *0.027* |
| Glycocholic_acid | 6522.50 | 4321.00 | 585.00* | *0.012* |
| Guanidine | 2604.58 | 2500.60 | 1.711^ | 0.091 |
| Hippuric_acid | 5720.00 | 2684.00 | 375.00* | *<0.001* |
| Homoveratric_acid | 1199.00 | 1260.00 | 905.00* | 0.961 |

*(Continued)*

**Table 2.** (*Continued*)

| Variable | Severity of Covid-19 | | Test statistic | p-value |
|---|---|---|---|---|
| | Asymptomatic/Mild | Severe | | |
| Hypoxanthine | 4269.00 | 928.00 | 361.00* | *<0.001* |
| Indole | 72516.85 | 33425.70 | 13.506^ | *<0.001* |
| Indoleacetic_acid | 3941.00 | 2386.00 | 379.50* | *<0.001* |
| Indolelactic_acid | 7088.00 | 3663.00 | 245.00* | *<0.001* |
| Inosinic_acid | 1951.00 | 409.00 | 239.50* | *<0.001* |
| Isobutyric_acid | 9089.00 | 9996.00 | 1220.00* | *0.005* |
| Isovalerylcarnitine | 28435.00 | 22621.00 | 723.00* | 0.119 |
| Kynurenic_acid | 2569.00 | 1838.00 | 596.00* | *0.007* |
| L_Acetylcarnitine | 91950.63 | 80032.73 | 1.252^ | 0.215 |
| L_Arginine | 5095.65 | 2078.80 | 9.214^ | *<0.001* |
| L_Carnitine | 36137.23 | 28644.60 | 3.181^ | *0.002* |
| L_Glutamine | 1151.00 | 1369.00 | 927.00* | 0.674 |
| L_Histidine | 2266.23 | 1702.71 | 8.643^ | *<0.001* |
| L_Kynurenine | 10672.00 | 10145.00 | 829.50* | 0.535 |
| L_Methionine | 6458.98 | 4043.20 | 5.643^ | *<0.001* |
| L_Norleucine | 23253.90 | 14443.91 | 7.143^ | *<0.001* |
| L_Phenylalanine | 290074.63 | 253585.6 | 2.282^ | *0.026* |
| L_Proline | 4493.00 | 3729.00 | 558.50* | *0.003* |
| L_Tryptophan | 306889.65 | 159829.4 | 11.566^ | *<0.001* |
| L_Valine | 9072.00 | 8366.00 | 484.00* | *<0.001* |
| m_Coumaric_acid | 31478.08 | 19619.36 | 6.053^ | *<0.001* |
| N_Acetylputrescine | 2481.00 | 2894.00 | 1017.00* | 0.220 |
| N_Acetylserotonin | 688.00 | 819.00 | 866.50* | 0.795 |
| N_Methylhydantoin | 12247.35 | 5606.98 | 11.224^ | *<0.001* |
| Niacinamide | 3058.00 | 1760.00 | 538.00* | *0.001* |
| Normetanephrine | 3216.83 | 2205.34 | 4.149^ | *<0.001* |
| Nutriacholic_acid | 15142.00 | 11152.00 | 542.00* | *0.002* |
| o_Tyrosine | 8467.05 | 14371.02 | -1.541^ | 0.128 |
| Oxalacetic_acid | 5210.08 | 4054.03 | 6.183^ | *0.001* |
| Oxypurinol | 1845.00 | 920.50 | 295.00* | *<0.001* |
| Pantothenic_acid | 6254.93 | 4607.49 | 2.756^ | *0.007* |
| Paracetamol_sulfate | 178.00 | 907.50 | 316.00* | *0.001* |
| Paraxanthine | 149056.08 | 38641.51 | 6.559^ | *<0.001* |
| PC_16_0_16_0 | 4229.82 | 4581.86 | -0.777^ | 0.440 |
| PC_18_1_9Z__18_1_9Z | 13135.03 | 10541.27 | 2.378^ | *0.020* |
| Phenylpropiolic_acid | 5721.33 | 3570.02 | 6.348^ | *<0.001* |
| Phosphoric_acid | 2211.45 | 1672.91 | 4.834^ | *<0.001* |
| Pipecolic_acid | 3300.28 | 2639.07 | 2.054^ | *0.043* |
| Propanal | 44206.00 | 46358.00 | 826.00* | 0.515 |
| Pyridoxal_5__phosphate | 2977.20 | 2906.02 | 0.919^ | 0.361 |
| Pyroglutamic_acid | 7546.23 | 19813.71 | -1.403^ | 0.167 |
| Sepiapterin | 4128.00 | 2592.00 | 555.00* | *0.005* |
| Serotonin | 11185.00 | 203.00 | 10.00* | *<0.001* |
| Sphingosine | 4111.25 | 4360.87 | -0.755^ | 0.453 |
| Succinic_acid | 1657.00 | 375.00 | 276.00* | *<0.001* |
| Succinylacetone | 1636.93 | 2115.98 | -2.720^ | *0.008* |

(*Continued*)

**Table 2.** (Continued)

| Variable | Severity of Covid-19 | | Test statistic | p-value |
|---|---|---|---|---|
| | Asymptomatic/Mild | Severe | | |
| Thyroxine | 1969.75 | 1525.43 | 2.294^ | *0.026* |
| Trimethylamine | 61600.93 | 43060.96 | 1.637^ | 0.105 |
| Urea | 2633.13 | 2462.13 | 1.594^ | 0.115 |
| Ureidosuccinic_acid | 2014.13 | 1793.20 | 1.938^ | 0.056 |
| Uric_acid | 37662.10 | 20716.82 | 4.215^ | *<0.001* |
| Uridine | 4276.95 | 3640.93 | 2.241^ | *0.028* |

*Mann Whitney U test to compare the median values.

^ Independent t-test to compare mean values.

citrate cycle, phenylalanine metabolism, phenylalanine, tyrosine, and tryptophan biosynthesis, pantothenate and coenzyme A biosynthesis, tryptophan, glycine, and serine (Fig 2A). Additionally, the same data set produced disease-enriched groups for Hartnup disease, acute seizures, critical illness (major trauma, severe septic shock, or cardiogenic shock), and hyperbaric oxygen exposure when it was searched against the "blood disease signatures database" (available in MetaboAnalyst 5.0 software). As further detailed in the text, the bulk of the diseases or conditions that emerged from this research have symptoms that are consistent with those listed among the most severe COVID-19 cases (Fig 2B).

## Determining Cut-off values for Biomarkers

To identify clinical biomarkers of disease severity, the ROC and AUC were performed for each clinical laboratory finding (CRP, ferritin, sTfR, LDH, etc.) and metabolite test. For the clinical laboratory findings, the highest accuracy level in predicting disease severity was for LDH (AUC = 1.000) followed by Ferritin (AUC = 0.988, 95% CI: 0.966 to 1.000), D-dimer (AUC = 0.936, 95% CI: 0.876 to 0.997), CRP (AUC = 0.904, 95% CI: 0.842 to 0.966), IL-6 (AUC = 0.919, 95% CI: 0.831 to 1.000), Hp (AUC = 0.792, 95% CI: 0.696 to 0.889), sTfR (AUC = 0.756, 95% CI: 0.653 to 0.858) and Neutrophils (AUC = 0.749, 95% CI: 0.643 to 0.854) (Table 3, Fig 3A). Hepcidin and Lymphocytes showed either insignificant or low AUC values (<0.70), indicating low accuracy in predicting COVID-19 severity. Optimal cut-off values for the laboratory findings as determined by Youden's index were 226 for LDH (SN = 100%, SP = 100%), 365 for Ferritin (SN = 95.6%, SP = 100%), 0.545 for D-dimer (SN = 93.0%, SP = 90.0%), 33.95 for IL-6 (SN = 87.1%, SP = 100%), 58.35 for CRP (SN = 73.3%, SP = 100%), 124.37 for Hp (SN = 79.5%, SP = 72.5%), 24.67 for sTfR (SN = 82.2%, SP = 60%), and 8.94 for Neutrophils (SN = 59.1%, SP = 85.0%) (Table 3).

Of all the 99 metabolite tests, five were identified as significant diagnostic tests in predicting COVID-19 disease severity; namely, K_4_Aminophenol (AUC = 0.883, 95% CI: 0.803 to 0.962), Acetaminophen (AUC = 0.949, 95% CI: 0.894 to 1.000), Acetaminophen glucuronide (AUC = 0.791, 95% CI: 0.539 to 1.000), Cytosine (AUC = 0.784, 95% CI: 0.680 to 0.887), and Paracetamol sulfate (AUC = 0.836, 95% CI: 0.660 to 1.000) (S1 Table, Fig 3B). Cut-off diagnostic value for K_4_Aminophenol was 381.5 (SN = 82.2%, SP = 90.0%), for Acetaminophen was 1595.5 (SN = 90.9%, SP = 91.2%), for Acetaminophen glucuronide was 1416.0 (SN = 78.0%, SP = 85.7%), for Cytosine was 818.0 (SN = 68.2%, SP = 90.0%), and for Paracetamol sulfate was 652.5 (SN = 81.0%, SP = 88.9%) (Table 3). Each biomarker was then dichotomized into two groups, low and high, based on its determined cut-off value.

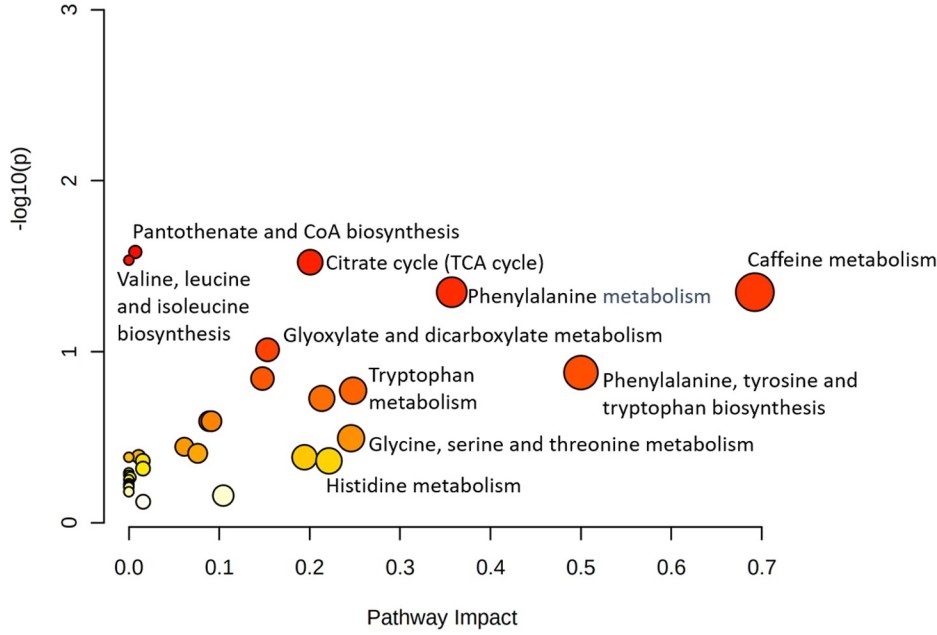

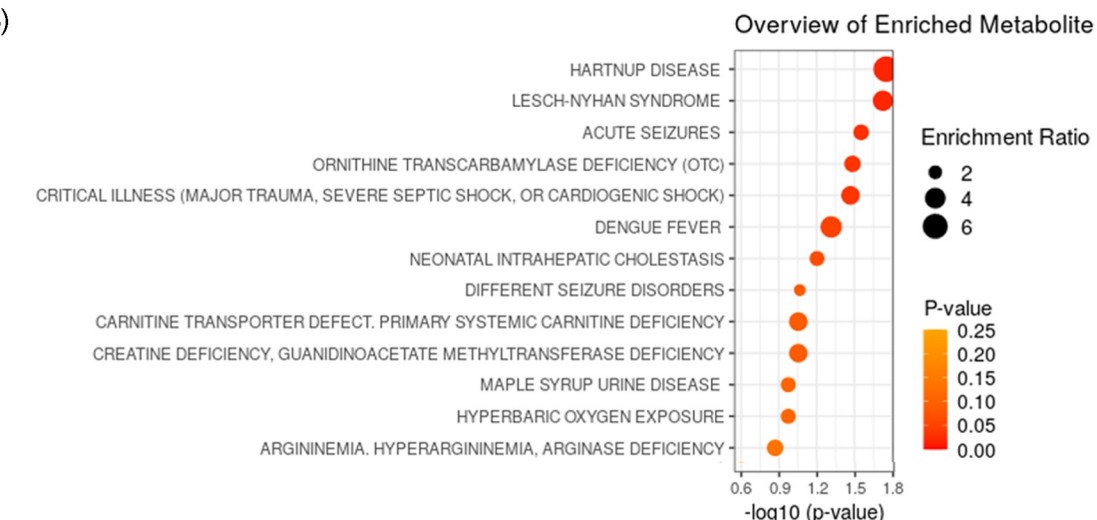

**Fig 2. Visualization of pathways enriched for significant altered metabolites (p<0.05) asymptomatic/mild versus severe COVID-19 using MetaboAnalyst pathway enrichment.** (A) Represent the enrichment pathway analysis against "small molecule pathway" database (SMPDB)" (found in the MetaboAnalyst 5.0 software) the results showed that the pathways for which the differentially abundant metabolites were most enriched were the citrate cycle, phenylalanine metabolism, phenylalanine, tyrosine, and tryptophan biosynthesis, pantothenate and coenzyme A biosynthesis, tryptophan. (B) Represent the enrichment pathway analysis against "blood disease signatures database," it generated disease-enriched groups for Hartnup disease, acute seizures, critical sickness (serious trauma, severe septic shock, or cardiogenic shock), and others (available in MetaboAnalyst 5.0 software). Nodes are coloured according to the level of significance for the enrichment (–log10(p)) and sized according to the number of associated dysregulated members (metabolites).

**Table 3. Statistical Significance of ROC AUC of clinical and metabolites tests in predicting severity of COVID-19.**

| No | Clinical test/ Metabolite | AUC | | P-value | Cut-off* | Sensitivity | | Specificity | |
|----|---------------------------|-------|----------------|---------|----------|-------|----------------|-------|----------------|
| | | Value | 95% CI:LL to UL | | | Value | 95% CI:LL to UL | Value | 95% CI:LL to UL |
| 1 | CRP | 0.904 | 0.842 to 0.966 | <0.001 | 58.35 | 73.3% | 57.8% to 84.9% | 100% | 89.1% to 100% |
| 2 | D—Dimer | 0.936 | 0.876 to 0.997 | <0.001 | 0.545 | 93.0% | 79.9% to 98.2% | 90.0% | 75.4% to 96.7% |
| 3 | Ferritin | 0.988 | 0.966 to 1.000 | <0.001 | 365 | 95.6% | 83.6% to 99.2% | 100% | 89.1% to 100% |
| 4 | IL-6 | 0.919 | 0.831 to 1.000 | <0.001 | 33.95 | 87.1% | 69.2% to 95.8% | 100% | 78.1% to 100% |
| 5 | LDH | 1.000 | 1.000 to 1.000 | <0.001 | 226 | 100% | 89.8% to 100% | 100% | 89.1% to 100% |
| 6 | Lymphocytes | 0.110 | 0.038 to 0.181 | <0.001 | - | | | | |
| 7 | Neutrophils | 0.749 | 0.643 to 0.854 | <0.001 | 8.94 | 59.1% | 43.3% to 73.3% | 85.0% | 69.5% to 93.8% |
| 8 | Hepcidin | 0.532 | 0.407 to 0.658 | 0.607 | - | | | | |
| 9 | Haptoglobin | 0.792 | 0.696 to 0.889 | <0.001 | 124.37 | 79.5% | 64.2% to 89.7% | 72.5% | 55.9% to 84.9% |
| 10 | Transferrin Receptor | 0.756 | 0.653 to 0.858 | <0.001 | 24.67 | 82.2% | 67.4% to 91.5% | 60.0% | 43.4% to 74.7% |
| 11 | K_4_Aminophenol | 0.883 | 0.803 to 0.962 | <0.001 | 381.5 | 82.2% | 67.4% to 91.5% | 90.0% | 75.4% to 96.7% |
| 12 | Acetaminophen | 0.949 | 0.894 to 1.000 | <0.001 | 1595.5 | 90.9% | 77.4% to 97.0% | 91.2% | 75.2% to 97.7% |
| 13 | Acetaminophen_glucuronide | 0.791 | 0.539 to 1.000 | 0.015 | 1416.0 | 78.0% | 62.0% to 88.9% | 85.7% | 42.0% to 99.2% |
| 14 | Cytosine | 0.784 | 0.680 to 0.887 | <0.001 | 818.0 | 68.2% | 52.2% to 80.9% | 90.0% | 75.4% to 96.7% |
| 15 | Paracetamol_sulfate | 0.836 | 0.660 to 1.000 | 0.002 | 652.5 | 81.0% | 65.4% to 90.9% | 88.9% | 50.7% to 99.4% |
| 16 | Model ^ | 0.996 | 0.989 to 1.000 | <0.001 | 3 | 100% | 89.1% to 100% | 92.5% | 78.5% to 98.0% |

*Optimal cut-off values were calculated only for tests showing high/moderate accuracy levels in predicting severity of COVID-19 (AUC > 0.70).

^Model includes six biomarkers: D-dimer, Ferritin, Neutrophils, Haptoglobin, Transferrin receptor, and Cytosine.

## Predicting COVID-19 disease severity

Predicting the severity of COVID-19 was done at two levels, first by using a single biomarker and then a combination of biomarkers. All clinical and metabolites tests were significantly and strongly associated with the severity of COVID-19. The proportion of patients with severe COVID-19 in the high group of each clinical/metabolites test was significantly higher than that in the low group. The strength of association, as measured by the odds ratio, between disease severity and the clinical and metabolites groups, was lowest for CRP (OR = 4.3, 95% CI: 2.6 to 7.1) and highest for D-dimer (OR = 120.0, 95% CI: 25.1 to 572.9) (Table 4).

A risk scoring system was developed to define a diagnostic model for COVID-19 disease severity based on a combination of important biomarkers used as predictors. All clinical laboratory findings and metabolites that were significantly associated with disease severity were considered as important biomarkers. After excluding biomarkers that were linearly related, and based on the statistical and clinical importance of all identified diagnostic biomarkers, we selected six predictors to conduct the risk-scoring predictive model. This model included the five lab findings (D-dimer, Ferritin, Neutrophils, Hp, and sTfR) and one metabolite (cytosine). A score was calculated for each patient. This score corresponded to the number of biomarkers that were of levels above their respective cut-off values (high group). The accuracy of predicting disease severity in the risk-scoring predictive model was reflected in a highly significant AUC value of 0.996 (95% CI: 0.989 to 1.000) (Fig 3C). The optimal cut-off risk score for the risk-scoring predictive model, as determined by Youden's Index, was 3 with a perfect sensitivity of 100% and a specificity of 92.5% (Table 3). Accordingly, all patients with high levels of at least three of the six predictors would be predicted as developing severe COVID-19 and none of the severe cases would be missed out.

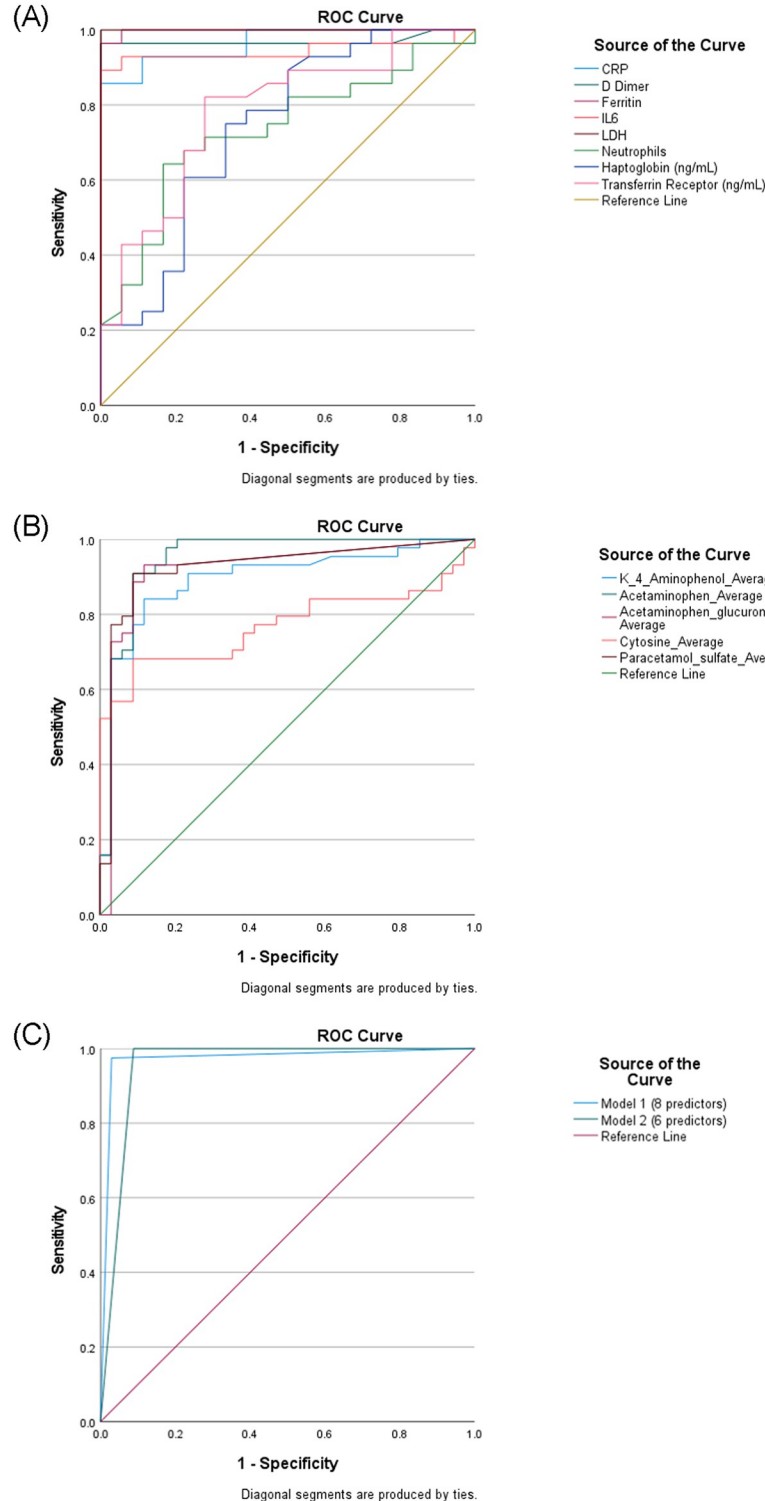

**Fig 3. ROC curves for prediction of COVID-19 severity based on patients' clinical and metabolites plasma levels.**
Clinical tests showing adequate AUC curves (>0.70) (A), metabolites showing adequate AUC values (>0.70), (B) and
ROC curves of predictive Model (C).

**Table 4. Cross-tabulations between severity of COVID-19 and clinical and metabolites tests.**

| Variables | | Mild/Asymptomatic | Severe | OR | 95% CI of OR | Chi-square | *P value* |
|---|---|---|---|---|---|---|---|
| Ferritin | Low | 40 (95.2%) | 2 (4.8%) | Reference | | 77.354 | <0.001 |
| | High | 0 (0.0%) | 43 (100.0%) | 20.8 | 5.4 to 83.3 | | |
| CRP | Low | 40 (76.9%) | 12 (23.1%) | Reference | | 47.949 | <0.001 |
| | High | 0 (0.0%) | 33 (100.0%) | 4.3 | 2.6 to 7.1 | | |
| D-dimer | Low | 36 (92.3%) | 3 (7.7%) | Reference | | 57.344 | <0.001 |
| | High | 4 (9.1%) | 40 (90.9%) | 120.0 | 25.1 to 572.9 | | |
| IL6 | Low | 18 (81.8%) | 4 (18.2%) | Reference | | 34.918 | <0.001 |
| | High | 0 (0.0%) | 27 (100.0%) | 5.5 | 2.3 to 13.3 | | |
| LDH | Low | 40 (100.0%) | 0 (0.0%) | Reference | | 83.000 | <0.001 |
| | High | 0 (0.0%) | 43 (100.0%) | - | – | | |
| Neutrophils | Low | 34 (65.4%) | 18 (34.6%) | Reference | | 17.272 | <0.001 |
| | High | 6 (18.8%) | 26 (81.3%) | 8.2 | 3.8 to 23.5 | | |
| Haptoglobin | Low | 29 (76.3%) | 9 (23.7%) | Reference | | 22.910 | <0.001 |
| | High | 11 (23.9%) | 35 (76.1%) | 10.3 | 3.7 to 28.1 | | |
| Transferrin receptor | Low | 24 (75.0%) | 8 (25.0%) | Reference | | 16.082 | <0.001 |
| | High | 16 (30.2%) | 37 (69.8%) | 6.9 | 2.6 to 18.7 | | |
| K_4_Aminophenol | Low | 36 (81.8%) | 8 (18.2%) | Reference | | 44.238 | <0.001 |
| | High | 4 (9.8%) | 37 (90.2%) | 41.6 | 11.5 to 150.5 | | |
| Acetaminophen | Low | 31 (88.6%) | 4 (11.4%) | Reference | | 52.242 | <0.001 |
| | High | 3 (7.0%) | 40 (93.0%) | 103.3 | 21.5 to 496.0 | | |
| Acetaminophen_glucuronide | Low | 6 (40.0%) | 9 (60.0%) | Reference | | 11.315 | <0.001 |
| | High | 1 (3.0%) | 32 (97.0%) | 21.3 | 2.3 to 200.9 | | |
| Cytosine | Low | 36 (72.0%) | 14 (28.0%) | Reference | | 29.439 | <0.001 |
| | High | 4 (11.8%) | 30 (88.2%) | 19.3 | 5.7 to 64.8 | | |
| Paracetamol_sulfate | Low | 8 (50.0%) | 8 (50.0%) | Reference | | 16.792 | <0.001 |
| | High | 1 (11.1%) | 34 (97.1%) | 34.0 | 3.7 to 312.1 | | |

Notes: OR–Odds Ratio; CI–Confidence Interval.

## Correlating ferritin with other laboratory findings and plasma metabolites

Among all patients, ferritin values were significantly correlated with the values of all other clinical tests except for hepcidin and sCD163. The significant correlations were all positive except for Lymphocytes that showed a moderate indirect correlation with ferritin (rho = - 0.520, p-value<0.001, 95% CI: -0.664 to -0.338). The strongest direct correlations of ferritin were found with LDH (rho = 0.785, p-value<0.001, 95% CI: 0.682 to 0.858), followed by D-dimer (rho = 0.630, p-value<0.001, 95% CI: 0.475 to 0.748) and CRP (rho = 0.618, p-value<0.001, 95% CI: 0.461 to 0.737); the weakest correlation was between ferritin and sTfR (rho = 0.282, p-value = 0.009, 95% CI: 0.067 to 0.472) (Table 5). Moreover, there were significant, mild to moderate indirect correlations between ferritin and several metabolites with serotonin showing the strongest indirect correlation (rho = -0.697, p-value<0.001, 95% CI: -0.836 to -0.474). Ferritin was also found to have a significant positive correlation with acetaminophen (rho = 0.670, p-value<0.001, 95% CI: 0.521 to 0.780), K_4_Aminophenol (rho = 0.573, p-value<0.001, 95% CI: 0.405 to 0.704) and cytosine (rho = 0.416, p-value<0.001, 95% CI: 0.215 to 0.583) (S2 Table).

**Table 5. Correlations between ferritin and the different clinical results.**

| No | Clinical test | n | Spearman Correlation coefficient (r) | P-value | 95% Confidence Interval LL to UL |
|----|---------------|---|--------------------------------------|---------|----------------------------------|
| 1 | CRP | 85 | 0.618 | <0.001 | 0.461 to 0.737 |
| 2 | D—dimer | 83 | 0.630 | <0.001 | 0.475 to 0.748 |
| 3 | IL 6 | 49 | 0.566 | <0.001 | 0.331 to 0.735 |
| 4 | LDH | 83 | 0.785 | <0.001 | 0.682 to 0.858 |
| 5 | Lymphocytes | 84 | -0.520 | <0.001 | -0.664 to -0.338 |
| 6 | Neutrophils | 84 | 0.370 | <0.001 | 0.163 to 0.546 |
| 7 | Hepcidin | 85 | 0.012 | 0.914 | -0.208 to 0.230 |
| 8 | Haptoglobin | 84 | 0.470 | <0.001 | 0.278 to 0.626 |
| 9 | Transferrin Receptor | 85 | 0.282 | 0.009 | 0.067 to 0.472 |
| 10 | sCD163 | 83 | -0.041 | 0.710 | -0.261 to 0.182 |

## Discussion

In this study, the concentration of several serum analytes and metabolites was measured in COVID-19 patients with no, mild or severe symptoms. Consistent with numerous previous studies, the serum concentration of analytes that are routinely measured in infected individuals including CRP, ferritin, IL-6, D-dimer, IL-6 and LDH was significantly elevated in patients with severe COVID-19 [22–26]. Also consistent with previous work was the observation that the levels of hepcidin [22,25], sTfR [22,23], and Hp [22,24] were either slightly-moderately elevated or not changed in COVID-19 patients with severe disease. Contrary to the suggestion of Zhou *et al* [25], our analysis showed that hepcidin is not a true predictor of disease severity in COVID-19 patients. Additionally, while data presented here show that the levels of sCD163 were reduced in severely ill patients, other studies have shown that sCD163 levels increase with disease severity [27]. This discrepancy could be a reflection of variations in methodology, sample collection timing, and/or differences in macrophage activity [28]. Irrespective of these discrepancies, variations in sCD163 concentration seem to have little, if any, impact on COVID-19 disease severity. Our data also showed that Hp phenotype distribution was similar in severe vs. asymptomatic/mild groups, which is in agreement with previous work which has suggested that Hp phenotype has no bearing on COVID-19 disease severity [29].

With regard to the metabolomics profiling of COVID-19 patients and as noted earlier, 60 metabolites decreased in the severe cases relative to asymptomatic/mild patients' group. The list of identified metabolites included several amino acids, vitamins and few fatty acids (Table 1). These are regarded as the fundamental elements that support the rise in cellular demands during illness. However, through catabolism pathways, they are also involved in innate and adaptive immune responses to infection [30,31]. Therefore, the decrease in some of the reported metabolites is consistent with previous studies, that reported lower levels of amino acids in hospitalized COVID-19 patients compared to asymptomatic ones [32,33]. The outcomes do in fact support the previously noted negative correlation between amino acids and immune responsiveness and hyper-inflammation indicators [34], which is characteristic of severe COVID 19. For example, the levels of Kynurenic acid in severe cases was found to be lower than that in asymptomatic/mild cases. Previous studies have suggested that the Tryptophan catabolite/ Kynurenine pathway may play a significant role in COVID-19 and critical COVID-19 [35]. Moreover, it appears that the increased level of kynurenine and the ratio of kynurenine to tryptophan is strongly correlated with the severity of COVID-19 patients [32,36,37]. Interestingly, the ratio of Kynurenic acid/Kynurenine did not significantly differ

between COVID-19 patients compared with non-COVID-19 controls, indicating no significant changes in Kynurenic acid activity, according to a systematic review [35]. This is indeed consistent with our finding that in patients with severe COVID-19, tryptophan and Kynurenic acid levels were significantly lower than in the counter group (Table 3).

Over the last three years, much time and effort has been spent on identifying serum biomarkers that could predict disease severity in COVID-19 patients with high accuracy. Elevated levels of serum biomarkers such as ferritin, IL-6, D-dimer and lactate dehydrogenase among others were reported to be valuable predictors of disease severity and death [22–26]. However, not all COVID-19 patients showing elevated levels in one or more of these biomarkers ended up with severe disease and death [38]. In other words, relying on one or more serum analytes tends to yield low prediction accuracy as evidenced by the fact that such approaches could only account for only a significant percentage of cases. In this context, numerous predictive models that relied on overlapping sets of variables drawn from COVID-19 patients' demographic data, clinical signs and symptoms, chest X-ray imaging and co-morbidities were proposed (reviewed in [39]. However, many of these severity predictive models suffer from a high degree of subjectivity and high likelihood of bias [39]. For example, a disease severity predictive model based on the static demographics (age, gender, occupation, urban vs. rural living, socio-economic status, profile, etc.) of >50000 Irish patients showed that modeling such parameter could predict hospitalization [(AUC 0.816 (95% CI 0.809, 0.822)], admission to ICU [AUC 0.885 (95% CI 0.88 0.89)] and death [AUC of 0.955 (95% CI 0.951 0.959)] [40]. In the same study, body mass index (BMI≥40) was shown to be a risk factor for ICU admission [OR 19.630] and death [OR 10.802]. Moreover, while rural living was found to associate with increased risk for hospitalization (OR 1.200 (95% CI 1.143–1.261)], urban living was found to associate with increased risk for ICU admission [OR 1.533 (95% CI 1.606–1.682)]. Another study which developed an artificial intelligence (AI)-based model based on 41 variables relating to patient's demographics, physical measurements, initial vital signs, comorbidities and laboratory findings in a cohort of 5628 Korean COVID-19 patients yielded a predictive power of >0.93 when 6 variables were used [40]. Besides the fact that this model could be skewed by the demographics components making it more population-specific than desired, achieving 93% accuracy by relying on 6 variables is cumbersome and difficult to apply in many poor countries and rural settings. Another AI-based model was developed by relying on laboratory findings including LDH, IL-6, D-dimer, fibrinogen, glucose, monokine induced by gamma interferon (MIG) and macrophage derived cytokine (MDC) levels in 60 COVID-19 Russian patients [41]. The model described by the study relied on eight parameters (creatinine, glucose, monocyte number, fibrinogen, MDC, MIG and CRP) to yield a predictive power of 83–87% [41]. In other words, this laboratory findings-based model failed to account for 13–17% of patients at risk of severe disease.

With the availability of six metabolite predictive biomarkers at our disposal, we sought to develop a high accuracy prediction model based on disparate data-integrating approaches; namely, patient laboratory findings and plasma metabolomics profiles. The statistical model developed and tested was based on ROC-AUC values. Although some metabolomes including acetaminophen, acetaminophen glucuronide and paracetamol sulfate significantly predicted COVID-19 severity, it was unlikely that these metabolomes were involved in the pathophysiology of COVID-19. Severe cases of the disease were more likely to receive more paracetamol than asymptomatic/mild cases. Therefore, these metabolomes were excluded from the predictive models. Accordingly, the predictive model was developed, and its prediction accuracy was internally validated using six biomarkers that were not linearly-related; namely, D-dimer, ferritin, neutrophil counts, Hp, sTfR and cytosine. Prediction accuracy of disease severity using this model was 0.996 (95% CI: 0.989 to 1.000) with optimal cut-off risk score of three

biomarkers. In other words, out of 100 patients with severe COVID-19 showing significant elevation in at least three of the six metabolites would predict disease severity in all 100 patients. The model has the advantage of yielding high predictive power with small set of variables (three laboratory findings) that can be easily and quickly acquired at minimal cost. Moreover, the predictive model can be dynamically applied independent of non-empirical clinical data (co-morbidities, signs and symptoms and loss of taste or smell among others) and can be dynamically applied as the disease progresses making timely and proper clinical interventions possible. That said, the utility of the model remains limited by the fact that the study was conducted retrospectively on a small number of samples. Another limitation in our study is that, with the number of COVID-19 cases gradually dwindling to almost zero in the UAE as in most parts of the world, we were not able to compile a new independent dataset with the same set of predictors as means of validating our prediction model. Future studies are recommended to test the validity of the suggested model on multiple datasets to ensure its generalizability.

## Conclusion

By integrating laboratory findings and metabolomic profiling data, a model to predict disease severity in COVID-19 patients was generated. The accuracy of the model was high (>98%), and it has the advantage of requiring three biomarkers to yield high sensitivity and specificity in predicting disease severity. The suggested model may prove useful in better managing COVID-19 patients at high risk of severe disease. Lastly, the fact that the model included cytosine as a biomarker and that cytosine is not usually included in routine laboratory testing for COVID-19 patients, merit further work on developing reliable and highly sensitive, yet quick and easy to perform, assays to measure serum cytosine concentration.

## Supporting information

**S1 Table. Statistical Significance of ROC AUC of metabolites in predicting severity of COVID-19.**
(DOCX)

**S2 Table. Correlations between Ferritin and the different metabolites.**
(DOCX)

## Author Contributions

**Conceptualization:** Nelson C. Soares, Mawieh Hamad.

**Data curation:** Nelson C. Soares, Amal Hussein, Jibran Sualeh Muhammad.

**Formal analysis:** Nelson C. Soares, Amal Hussein.

**Funding acquisition:** Nelson C. Soares, Mawieh Hamad.

**Investigation:** Nelson C. Soares, Jibran Sualeh Muhammad, Mohammad H. Semreen, Gehad ElGhazali, Mawieh Hamad.

**Methodology:** Nelson C. Soares, Jibran Sualeh Muhammad, Mohammad H. Semreen, Gehad ElGhazali, Mawieh Hamad.

**Project administration:** Mawieh Hamad.

**Resources:** Mawieh Hamad.

**Software:** Amal Hussein, Mawieh Hamad.

**Supervision:** Nelson C. Soares, Mawieh Hamad.

**Validation:** Nelson C. Soares, Mohammad H. Semreen, Mawieh Hamad.

**Visualization:** Nelson C. Soares, Amal Hussein, Mawieh Hamad.

**Writing – original draft:** Nelson C. Soares, Amal Hussein, Jibran Sualeh Muhammad, Mawieh Hamad.

**Writing – review & editing:** Nelson C. Soares, Amal Hussein, Mawieh Hamad.

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
