## [Decision Letter · Decision Letter 0]

6 Nov 2022

PONE-D-22-27539Plasma metabolomics profiling identifies new predictive biomarkers for disease severity in COVID-19 patientsPLOS ONE

Dear Dr. Hamad,

Thank you for submitting your manuscript to PLOS ONE. After careful consideration, we feel that it has merit but does not fully meet PLOS ONE’s publication criteria as it currently stands. Therefore, we invite you to submit a revised version of the manuscript that addresses the points raised during the review process.

We look forward to receiving your revised manuscript.

Kind regards,

Konlawij Trongtrakul, MD, PhD

Academic Editor

PLOS ONE

Journal Requirements:

2. Please ensure that you have specified (1) whether consent was informed and (2) what type you obtained (for instance, written or verbal, and if verbal, how it was documented and witnessed). If your study included minors, state whether you obtained consent from parents or guardians. If the need for consent was waived by the ethics committee, please include this information.

"This work was supported by research grant CoV19-0305 (MH), seed grant 2001110138, University of Sharjah, UAE. This research is part of the -Human Disease Biomarkers Discovery Research Group-study. The authors wish to acknowledge the generous support of the Research Institute for Medical and Health Sciences, University of Sharjah UAE."

"This work was supported by research grant CoV19-0305 (MH), seed grant 2001110138 (NCS), University of Sharjah, UAE. This research is part of the -Human Disease Biomarkers Discovery Research Group-study.  The authors wish to acknowledge the generous support of the Research Institute for Medical and Health Sciences, University of Sharjah UAE"

5. We note that Figure 1 in your submission contain copyrighted images. All PLOS content is published under the Creative Commons Attribution License (CC BY 4.0), which means that the manuscript, images, and Supporting Information files will be freely available online, and any third party is permitted to access, download, copy, distribute, and use these materials in any way, even commercially, with proper attribution. For more information, see our copyright guidelines: http://journals.plos.org/plosone/s/licenses-and-copyright.

Additional Editor Comments:

The manuscript provides an interesting issue in determining COVID-19 severity using the patient's laboratory investigations combined with significant metabolites from LC-MS/MS. The models provided by the authors seem optimistic, with AUC nearly equal to 1. Unfortunately, there is a concern about needing more statistical analysis. I recommend performing multivariable logistic regression analysis or binary regression analysis as it is more appropriate to finalize the model. Afterward, the author can test the AUC of the model. Nonetheless, a suitable criterion for selecting metabolites that reflect more severe disease in the model is necessary, rather than including acetaminophen or related medication.

Reviewers' comments:

Reviewer's Responses to Questions

**Comments to the Author**

1. Is the manuscript technically sound, and do the data support the conclusions?

Reviewer #1: Partly

Reviewer #2: Partly

2. Has the statistical analysis been performed appropriately and rigorously? 

Reviewer #1: Yes

Reviewer #2: N/A

3. Have the authors made all data underlying the findings in their manuscript fully available?

Reviewer #1: Yes

Reviewer #2: No

4. Is the manuscript presented in an intelligible fashion and written in standard English?

Reviewer #1: Yes

Reviewer #2: Yes

5. Review Comments to the Author

Reviewer #1: The authors developed the predictive models of COVID-19 severity based upon the previously common biomarkers together with plasma metabolomes. I have a few questions and suggestions as of the following:

1. The authors should give more details regarding sample collection and treatment history of the enrolled cases, including 1) the blood samples were collected at which day of their symptoms (please give the range), 2) what was the treatment for symptomatic patients (both mild and severe cases)?, and 3) did the patients receive treatment (by physicians or by somebody else) prior to the sample collection? In fact, the treatment and the onset of disease at the time of sample collection may confound the predictive models.

2. Were there any patients who developed the progression of symptoms (from mild to severe) and was there mortality among the severe cases? If there were, the authors should identify the differences of biomarkers and plasma metabolomes between the progressive cases vs. the non-progressive cases and between the survival vs. the non-survival cases. This may contribute to the identification of prognostic markers in this study as well.

3. Besides age, sex, and co-morbidities, did the authors include some demographic information of the patients in the predictive models? For example, body mass index, smoking history, alcohol drinking history, physical activity, and medications since these can affect pulmonary and immune function that may determine the disease severity of COVID-19. In other words, adding these parameters to the models may increase the accuracy of prediction.

4. Please give the rationale why the authors ran only the positive mode on the LC/MS. According to my previous run, the negative mode of this LC/MS condition could provide a lot of additional endogenous metabolomes, especially free fatty acids and phospholipids that play a crucial role in the pathophysiology of several diseases.

5. Considering the positive mode LC/MS used in this study, this protocol is robust for the determination of acylcarnitines and phospholipids in human plasma. Again, these endogenous metabolomes play an important role in pathophysiology of several diseases. However, most of these metabolomes were not included in the 99 identified metabolomes of this study. Therefore, the authors should quantify these metabolomes and add them to the models.

6. Among 99 identified metabolomes, some of them are considered exogenous compounds, such as caffeine, chlorpheniramine, acetaminophen, acetaminophen glucuronide, and paracetamol sulfate. It is less likely that these metabolomes are involved in the pathophysiology of COVID-19, and hence I do not think that they should be included in the predictive models. Specifically, the authors’ results showed that acetaminophen, acetaminophen glucuronide, and paracetamol sulfate were 3 of 5 significant metabolomes that predicted COVID-19 severity. According to these findings, it was likely that the severe cases received more paracetamol than asymptomatic/mild cases. Therefore, the exogenous metabolomes should be reported only in terms of the difference between groups of patients, but they should be excluded from the predictive models.

7. The authors demonstrated that LDL exerted the highest accuracy level in predicting COVID-19 severity among the clinical biomarkers (Page 13, Lines 285-286). This finding was really interesting and should be included in the discussion section.

8. Since the authors showed that LDL exhibited that highest accuracy level in predicting COVID-19 severity, it was likely that lipid status and insulin sensitivity play a critical role in severity of COVID-19. Did the authors include triglyceride, total cholesterol, HDL, VLDL, glucose, insulin, and HOMA-IR in the models? If the author did, what were the results? If the authors did not, I suggest that the authors should include them in the models.

Reviewer #2: I have 4 major comments, added here. The remaining will be uploaded as a document. From my perspective those 4 points are very important blockers to a publication at this stage.

1. As briefly mentioned at the end of the discussion session this study requires a validation. No test / validation population was curved out from the initial cohort and all reported results were calculated on the full set of data. As such they are unreliably optimistic. This is hinted by the unusually high AUC values. I would strongly recommend that the proposed model(s) performance is validated in a new cohort prior to publication.

2. As only few metabolites resulting from LC-MS were retained for the models I recommend that prior to publication the identification of those is validated against a standard. This will give the high confidence level (MSI 1) required to be of use to the scientific community. More specifically in the case of COVID-19 patients a lot of cytosine containing compounds are elevated. For many of those compounds the cytosine sub-structure tends to break easily at the MS source resulting in multitude of cytosine events in an acquired TIC. Each of those events will have a good match to cytosine based on MS/MS, but not on RT. Please make sure that the metabolic feature you selected for you model overlap in RT with a cytosine standard for your method i.e., column and LC-MS conditions.

3. I strongly believe that predictive models should not rely on detecting medication (K_4_aminophenol, acetaminophen) in patient’s blood. This is highly subjective to patient’s lifestyle choices and hospital practices.

4. The study states in the abstract that “almost all such models, which relied on serum/plasma biomarkers, clinical data or a combination of both, were subsequently deemed as cumbersome, inadequate and/or subject to bias”. However, very few predictive models of COVID severity were discussed in this work. I don’t believe that this statement was well defended. A more thorough discussion on the alignment / misalignment of this study findings with previous models will be beneficial to the community. The LC-MS findings more specifically were lacking coverage.

6. PLOS authors have the option to publish the peer review history of their article (what does this mean?). If published, this will include your full peer review and any attached files.

Reviewer #1: No

Reviewer #2: No

---

## [Author Response · Author response to Decision Letter 0]

26 May 2023

Reviewer #1 (R1): The authors developed the predictive models of COVID-19 severity based upon the previously common biomarkers together with plasma metabolomes. I have a few questions and suggestions as of the following:

Q1. The authors should give more details regarding sample collection and treatment history of the enrolled cases, including 1) the blood samples were collected at which day of their symptoms (please give the range), 2) what was the treatment for symptomatic patients (both mild and severe cases)?, and 3) did the patients receive treatment (by physicians or by somebody else) prior to the sample collection? In fact, the treatment and the onset of disease at the time of sample collection may confound the predictive models.

Response (R1 Q1): We thank the reviewer for his/her suggest. More details were provided regarding sample collection as per the reviewer’s suggestion. Regarding the issue of patient management, please note that this was in the early stages of the pandemic when treatment options were very limited and treatment guidelines were constantly changing. Please see Material and Methods, sample collection, page (P) 6-7 Line (L) 131-134. 

Q2. Were there any patients who developed the progression of symptoms (from mild to severe) and was there mortality among the severe cases? If there were, the authors should identify the differences of biomarkers and plasma metabolomes between the progressive cases vs. the non-progressive cases and between the survival vs. the non-survival cases. This may contribute to the identification of prognostic markers in this study as well.

Response (R1 Q2): While the reviewer’s comment is very relevant, patient follow up was not possible Aas this was a retrospective study.  

Q.3. Besides age, sex, and co-morbidities, did the authors include some demographic information of the patients in the predictive models? For example, body mass index, smoking history, alcohol drinking history, physical activity, and medications since these can affect pulmonary and immune function that may determine the disease severity of COVID-19. In other words, adding these parameters to the models may increase the accuracy of prediction.

Response (R1 Q3): We thank the reviewer for his/her comment and we totally agree with the reviewer’s comment; smoking history, alcohol consumption, etc. do indeed complicate the clinical picture and can be useful in predicating disease progression. In fact, previous models have evaluated the utility of some such demographic variables and showed that they are indeed very useful. (please see Boudou, M., et al. Sci Rep 11, 18474 (2021). https://doi.org/10.1038/s41598-021-98008-6). In this study however, no demographic parameters were included as the focus of the study was to test whether combining key routine lab findings such as plasma ferritin levels plus differential metabolomics profiling can be used to predict patient severity. To accommodate the reviewer’s concern, the reasoning behind excluding demographic parameters was briefly described. Please see P11 L39-37 

Q4. Please give the rationale why the authors ran only the positive mode on the LC/MS. According to my previous run, the negative mode of this LC/MS condition could provide a lot of additional endogenous metabolomes, especially free fatty acids and phospholipids that play a crucial role in the pathophysiology of several diseases.

Response (R1 Q4). We agree with the reviewer on the ionization mode of the processed samples; however, the goal of the study was metabolic profiling against our data base (HMDB), which is not comprehensive for metabolites prone to ionization in negative mode, such as fatty acids and phospholipids; thus, switching to negative mode will not improve the identification of certain metabolites. Furthermore, we have previously conducted studies with negative mode, and the results were not satisfactory. 

Q5. Considering the positive mode LC/MS used in this study, this protocol is robust for the determination of acylcarnitines and phospholipids in human plasma. Again, these endogenous metabolomes play an important role in pathophysiology of several diseases. However, most of these metabolomes were not included in the 99 identified metabolomes of this study. Therefore, the authors should quantify these metabolomes and add them to the models.

Response (R1 Q5). Typically, the formation of acylcarnitine (carriers of the acetyl group) as an intermediate will occur in the intermembranous space of the mitochondria, transported into the matrix of the mitochondria, then hydrolyzed into carnitine and Acyl-CoA, so the time frame of its availability is short, and therefore, is identified as carnitine.

 

Q6. Among 99 identified metabolomes, some of them are considered exogenous compounds, such as caffeine, chlorpheniramine, acetaminophen, acetaminophen glucuronide, and paracetamol sulfate. It is less likely that these metabolomes are involved in the pathophysiology of COVID-19, and hence I do not think that they should be included in the predictive models. Specifically, the authors’ results showed that acetaminophen, acetaminophen glucuronide, and paracetamol sulfate were 3 of 5 significant metabolomes that predicted COVID-19 severity. According to these findings, it was likely that the severe cases received more paracetamol than asymptomatic/mild cases. Therefore, the exogenous metabolomes should be reported only in terms of the difference between groups of patients, but they should be excluded from the predictive models.

Response (R1 Q6): The authors thank the reviewer for this important comment and totally agree him/her on excluding the exogenous metabolomes from the predictive models. Accordingly, the authors decided to remove Model 1 which includes K-4-aminophenol and acetaminophen as predictors. The manuscript now reports a single model for predicting disease severity. The reported model includes a total of six predictors (5 lab findings and one metabolite, cytosine). Edits are highlighted in the manuscript in the subsection entitled “Predicting COVID-19 disease severity’ under the Results section. Furthermore, under the Discussion section, a few statements were added to explain why these metabolomes were excluded from the predictive model despite showing significant predictive findings in their ROC-AUC values. All edits and additions were highlighted in yellow in the Results, Discussion and Conclusion sections. 

Q7. The authors demonstrated that LDL exerted the highest accuracy level in predicting COVID-19 severity among the clinical biomarkers (Page 13, Lines 285-286). This finding was really interesting and should be included in the discussion section.

Response (R1 Q7): We thank the reviewer for drawing our attention to this point. Unfortunately, the term LDL was erroneously used a couple of times in place of the correct term (LDH); we sincerely apologize for this error.

Q8. Since the authors showed that LDL exhibited that highest accuracy level in predicting COVID-19 severity, it was likely that lipid status and insulin sensitivity play a critical role in severity of COVID-19. Did the authors include triglyceride, total cholesterol, HDL, VLDL, glucose, insulin, and HOMA-IR in the models? If the author did, what were the results? If the authors did not, I suggest that the authors should include them in the models.

Response (R1 Q8): The authors thank the reviewer. Unfortunately, lipid status and insulin sensitivity were not included in the patients’ dataset. This point was addressed as a limitation in the discussion.

 

Reviewer #2: I have 4 major comments, added here. The remaining will be uploaded as a document. From my perspective those 4 points are very important blockers to a publication at this stage.

Q1. As briefly mentioned at the end of the discussion session this study requires a validation. No test / validation population was curved out from the initial cohort and all reported results were calculated on the full set of data. As such they are unreliably optimistic. This is hinted by the unusually high AUC values. I would strongly recommend that the proposed model(s) performance is validated in a new cohort prior to publication.

Response (R2 Q1): The authors agree to this point and have addressed it as a limitation under the Discussion section. P22 L455-459

2. As only few metabolites resulting from LC-MS were retained for the models I recommend that prior to publication the identification of those is validated against a standard. This will give the high confidence level (MSI 1) required to be of use to the scientific community. More specifically in the case of COVID-19 patients a lot of cytosine containing compounds are elevated. For many of those compounds the cytosine sub-structure tends to break easily at the MS source resulting in multitude of cytosine events in an acquired TIC. Each of those events will have a good match to cytosine based on MS/MS, but not on RT. Please make sure that the metabolic feature you selected for you model overlap in RT with a cytosine standard for your method i.e., column and LC-MS conditions.

Response (R2 Q2): For untargeted metabolomics analysis we follow a standard protocol according to the manufacturer’s recommendations “Acquisition of high quality LC-MS/MS data follows sample preparation in non-targeted metabolomics workflows, with the T-ReX® LC-QTOF solution, no LC-MS/MS parameter optimization is required” and “To analyze large sample cohorts which require high retention time stability the Elute UHPLC in combination with the dedicated T-ReX® Elute Metabolomics-kit: RP was used”. The Reversed-Phase LC column kit enables matching of retention times to values in the Bruker HMDB Metabolite Library. 

 So, we use the same column, Elute UHPLC System, mobile phase gradients, and all parameters and settings. 

The m/z measurements were externally calibrated using 10 mM of sodium formate before sample analysis. In addition, sodium formate solution was injected at the beginning of each sample run and used for internal calibration during data processing.

TRX-2101/RT-28-calibrants for Bruker T-ReX LC-QTOF (Nova Medical Testing Inc.) was injected before sample analysis to check and test the performance of the column reversed-phase liquid chromatography (RPLC) separation, multipoint retention time calibration, and the mass spectrometer. Also, TRX-3112-R/MS Certified Human serum for Bruker T-ReX LC-QTOF solution (Nova Medical Testing Inc.) was prepared from pooled human blood and injected before sample analysis to check the performance of the LC-MS instruments. 

As a standard protocol, the test mixture data files were uploaded to Metaboscape 4.0 and confirmed the presence of the entire set of metabolites in the samples by choosing the set with a higher annotation quality score (AQ score) representing the best retention time values, MS/MS score, m/z values, mSigma.

Based on the nearest retention time values registered in the HMDB library all metabolites are filtered

Q3. I strongly believe that predictive models should not rely on detecting medication (K_4_aminophenol, acetaminophen) in patient’s blood. This is highly subjective to patient’s lifestyle choices and hospital practices.

Response (R2 Q3): The authors thank the reviewer for this important comment and totally agree about on excluding the exogenous metabolomes from the predictive models. Accordingly, the authors decided to remove Model 1 which includes K-4-aminophenol and acetaminophen as predictors. The manuscript now reports a single model for predicting disease severity. The reported model includes a total of six predictors (5 lab findings and one metabolite, cytosine). Edits are highlighted in the manuscript in the subsection entitled “Predicting COVID-19 disease severity’ under the Results section. Furthermore, under the Discussion section, a few statements were added to explain why these metabolomes were excluded from the predictive model despite showing significant predictive findings in their ROC-AUC values. All edits and additions were highlighted in yellow in the Results, Discussion and Conclusion sections. 

Q4. The study states in the abstract that “almost all such models, which relied on serum/plasma biomarkers, clinical data or a combination of both, were subsequently deemed as cumbersome, inadequate and/or subject to bias”. However, very few predictive models of COVID severity were discussed in this work. I don’t believe that this statement was well defended. A more thorough discussion on the alignment / misalignment of this study findings with previous models will be beneficial to the community. 

Response (R2 Q4): We thank the reviewer for his/her comment. Firstly, the reviewer’s comment made it clear to us that we perhaps overstated the case regarding the lack of utility of previous models which relied on lab findings, clinical symptoms and demographics. Our commentary on previous models both in the abstract and in the discussion were watered down not to overstate the case. Secondly, as the reviewer suggest, the discussion was expanded and re-organized to better argue our case . Please see Abstract, lines 2-4 and discussion section, page … , line … 

Q5 The LC-MS findings more specifically were lacking coverage.

Response (R2 Q5): We understand the reviewer's criticism regarding the identification of specific metabolites; nevertheless, the identification was compared to the 800 metabolites present in our HMDB, therefore we chose to use a more stringent set of criteria as described in M&M and keep those metabolites assigned by MS/MS only. This reduced the risk of false positive identifications. Accordingly, the number of metabolites here reported is comparable to other studies using the similar workflow

---

## [Decision Letter · Decision Letter 1]

26 Jul 2023

Plasma metabolomics profiling identifies new predictive biomarkers for disease severity in COVID-19 patients

PONE-D-22-27539R1

Dear Dr. Hamad,

We’re pleased to inform you that your manuscript has been judged scientifically suitable for publication and will be formally accepted for publication once it meets all outstanding technical requirements.

Kind regards,

Konlawij Trongtrakul, MD PhD

Academic Editor

PLOS ONE

Additional Editor Comments (optional):

Reviewers' comments:

Reviewer's Responses to Questions

**Comments to the Author**

1. If the authors have adequately addressed your comments raised in a previous round of review and you feel that this manuscript is now acceptable for publication, you may indicate that here to bypass the “Comments to the Author” section, enter your conflict of interest statement in the “Confidential to Editor” section, and submit your "Accept" recommendation.

Reviewer #1: (No Response)

2. Is the manuscript technically sound, and do the data support the conclusions?

Reviewer #1: Yes

3. Has the statistical analysis been performed appropriately and rigorously? 

Reviewer #1: Yes

4. Have the authors made all data underlying the findings in their manuscript fully available?

Reviewer #1: Yes

5. Is the manuscript presented in an intelligible fashion and written in standard English?

Reviewer #1: Yes

6. Review Comments to the Author

Reviewer #1: (No Response)

7. PLOS authors have the option to publish the peer review history of their article (what does this mean?). If published, this will include your full peer review and any attached files.

Reviewer #1: No

---

## [Editor Report · Acceptance letter]

3 Aug 2023

PONE-D-22-27539R1 

Plasma metabolomics profiling identifies new predictive biomarkers for disease severity in COVID-19 patients 

Dear Dr. Hamad:

I'm pleased to inform you that your manuscript has been deemed suitable for publication in PLOS ONE. Congratulations! Your manuscript is now with our production department. 

Kind regards, 

on behalf of

Konlawij Trongtrakul 

Academic Editor

PLOS ONE